# Neural Graph Modelling of Whole Slide Images for Survival Analysis

**Callum Mackenzie**
University of Warwick, UK
c.mackenzie1@warwick.ac.uk

**Muhammad Dawood**
University of Warwick, UK
muhammad.dawood@warwick.ac.uk

**Simon Graham**
University of Warwick, UK
simon.graham@warwick.ac.uk

**Mark Eastwood**
University of Warwick, UK
mark.eastwood@warwick.ac.uk

**Fayyaz Minhas**
University of Warwick
fayyaz.minhas@warwick.ac.uk

## Abstract

Evaluation of a cancer patient's prognostic outlook is an essential step in the clinical decision-making process, involving the assessment of complex tissue structures in multi-gigapixel whole slide images (WSIs). Effective risk stratification of patients from WSIs has proven challenging despite several approaches across the literature due to their large size and inability of existing approaches to effectively model inter-relationships between different tissue components. We propose a graph neural network (GNN) model that performs pairwise ranking of graph representations of WSIs based on survival scores. The proposed approach translates spatially-localised deep features along with their spatial context to a graph neural network to produce survival scores. Analysis over breast cancer patients from The Cancer Genome Atlas (TCGA) shows that the proposed GNN approach is able to rank patients with respect to their disease-specific survival times with a concordance index of $0.672 \pm 0.058$. This is a significant improvement over existing state of the art and paves the way for neural graph modelling of WSI data for survival prediction for other cancer types.

## 1 Introduction

Breast cancer (BCa) most commonly affects women, with around 55,500 women per year diagnosed in the UK and 1 in 7 women will develop breast cancer in their lifetime [1]. As of 2022 breast cancer is the most commonly diagnosed cancer in the UK and is the second most common form of cancer related deaths in women. According to Public Health England, 85% of women diagnosed with breast cancer in England survive their disease for five years or more [2], with survival rates doubling over the last 40 years driven by more thorough early cancer detection and improved treatment regimes. Mortality rates are projected to fall by a further 26% before 2035.

In the process of an individual's breast cancer diagnosis, a tissue sample is typically taken from the tumour region and analysed by a histopathologist. The sample is stained using Haematoxylin and Eosin (H&E) and then digitised using a high resolution whole slide imaging scanner resulting in a multi-gigapixel ( $100,000 \times 100,000$ pixels at 0.25 microns per pixel resolution) whole slide image (WSI). Pathologist assessment of the WSI involves inspection of the morphological features of the tissue cells to draw conclusions relating to the grade, type, and hormone receptor status of the patient's cancer. All of these features have strong correlations to a given patient's likelihood of survival and is used to determine treatment options.

C. Mackenzie et al., Neural Graph Modelling of Whole Slide Images for Survival Analysis. *Proceedings of the First Learning on Graphs Conference (LoG 2022)*, PMLR 198, Virtual Event, December 9–12, 2022.

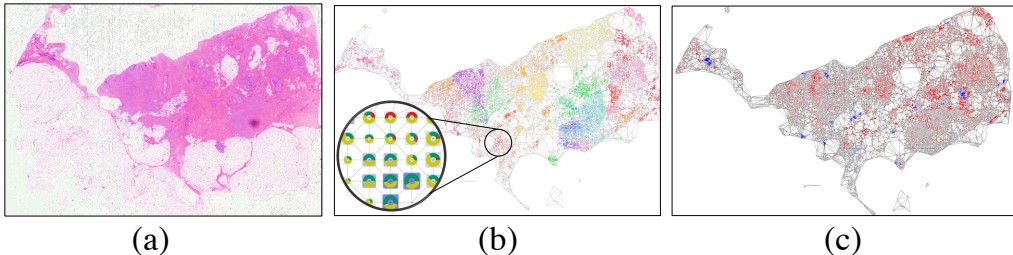

**Figure 1:** Major steps in the proposed approach: (a) Input Whole Slide Image. (b) Patch level feature extraction and graph construction by spatial clustering and inter-linking neighbouring patches. (c) Using Graph Neural Network based node level survival scoring to generate both node level and WSI level predictions.

Currently the analysis of WSIs is performed by human histopathologists and there is a significant possibility for human error and bias in making decisions. Most notable in this regard is the assesment of nuclear pleaomorphism, a component step of the Nottingham grading system. This assessment is often the most subjective and so pathologists differ markedly in their nuclear grading. It has been found that breast specialists will assign higher grades than non-specialists [3] based of the assessment of the tissue nuclear pleaomorphism. This degree of subjectivity is not conducive to reproducible and reliable categorisations. There is, therefore, significant motivation to create an objective approach method for survival prediction that avoids human subjectivity by stratifying patients based on their risk.

Weakly supervised learning from whole slide images has seen a number of contributions recently, particularly aiming to solve classification tasks. However, the regression based problem of survival prediction is considerably more difficult as disease risk presents itself as a combination of a number of histological features which correspond to disease status. While there has been some work done aiming to predict survival from local information extracted from sub-patches of the WSI, there are only a few frameworks that perform prediction at a global WSI level. Existing methods [4, 5] follow a structure of taking a large number of random patches from the image and clustering them according to a rule set, commonly by phenotype. These clusters at patch level will undergo aggregation and produce a prediction. Alternative methods include MCAT, a co-attention mapping between WSIs and genomic sequencing formulated in an embedding space [6]. A number of studies have also investigated model performance when patient data is also fed in however the goal of this work is to construct a model capable of WSI level survival prediction alone. Furthermore, there have been successful integrations when using a NTL (Nucleus, Tumour, and Lymphocyte) data along side the RGB WSI [7]. However, the concordance between predicted and true survival using whole slide image data remains low.

A considerable limitation to the existing systems in the literature is the loss of spatial context between the random patches. Macro-scale histological structures formed from specific cell types contribute heavily to a prognostic prediction for a patient. Furthermore, existing implementations have also relied on external information (such as gene expression patterns) from the WSI to boost scores. The motivation for this work is to learn survival from the WSI alone.

In order to identify significant survival associated features in multi-gigapixel WSIs, we need an effective way of capturing the inter-relationship between different components in the WSI. For this purpose, graph based modelling of whole slide images is an attractive solution. Existing work in this domain (SlideGraph and SlideGraph+) [8, 9] has shown that it is possible to solve classification problems for breast cancer receptor status prediction using WSI-level graph neural networks. In this work, we demonstrate the effectiveness of graph based neural models of whole slide images for survival analysis by using a pairwise ranking loss over predicted survival scores. The proposed approach results in significant improvement over existing state of the art approaches in this domain and paves the way for more effective graph based models.

## 2 Methodology

### 2.1 Whole slide image and survival data

We collected 1,133 whole slide images of Formalin-Fixed paraffin-Embedded (FFPE) Hematoxylin and Eosin (H&E) stained tissue section of 1084 breast cancer patients from The Cancer Genome Atlas (TCGA) [10, 11]. The Disease Specific Survival (DSS) data for these patients were collected from the TCGA Pan-Cancer Clinical Data Resource (TCGA-CDR) [12]. DSS is the length of time between the data being taken and a disease specific event occurring, in this case death. For some of the patients the survival data was missing and, consequently, was not used in the study. In line with clinical practice, patient survival times were censored at 10 years.

### 2.2 Pre-processing

Quality of a digitised H&E stained tissue section can be adversely affected by tissue processing artefacts such as tissue folds and pen-marking originating from the histology laboratory. We filter these artefacts by segmenting the tissue region of whole slide image and ignoring regions with tissue artefacts (tissue folding, and pen-marking etc.) using our in-house tissue segmentation model. As an entire WSI at full resolution can be of size $100,000 \times 150,000$ pixels and can be challenging to fit into a GPU memory, therefore we tile each WSI into patches of size $512 \times 512$ pixels at a spatial resolution of 0.25 microns-per-pixel (MPP). Patches with less than 40% of tissue region (mean pixel intensity of 60% pixels being higher than 200) are discarded and the rest of patches (tumor and non-tumor) are used in the study. The total number of $512 \times 512$ patches in the data is $8,487,768$.

### 2.3 Graph Representation of a Whole Slide Image

For each WSI, we construct its graph representation, $G_i = G(\mathbf{X_i})$ where $\mathbf{X_i}$ is the set of features extracted from each patch within the WSI. In general, a WSI is represented as a set $X = \{\mathbf{p}_m | m = 1, ..., M\}$ where $\mathbf{p}_m \equiv (\mathbf{x}_m, \mathbf{f}_m)$ is composed of the location of the given patch, $\mathbf{x}_m$, along with its feature vector, $\mathbf{f}_m$. The graph construction process can be broken down into three steps which are explained below (see Figure-1):

#### 2.3.1 Patch-Level Feature Extraction

The tissue region in the whole slide image is broken down into patches and representative features of each patch are then extracted. Specifically, we extract Shuffle-Net [13] based deep features. We encode the patch image of size $512 \times 512$ pixels into a 1024-dimensional feature vector by extracting latent representation of the penultimate fully-connected layer in Shuffle-Net. However, other types of feature representations can be used in the proposed framework as well such as patch level cellular composition [14].

#### 2.3.2 Spatial Clustering

Due to the large size of a whole slide image and the large number of tissue patches, it is necessary to reduce the size of the graph while maintaining as much of the stored information as possible. This is done to reduce the computational cost of learning and subsequent analysis with the graphs. In line with the SlideGraph approach, agglomerative clustering [15] is used to group patches in the original set $P$ into $K$ clusters represented by the set $C = \{c_k | k = 1, ..., K\}$ based on spatial neighbourhood and feature similarity. The number of clusters is different for each WSI depending upon its size and tissue heterogeneity. For further details, the interested reader is referred to [8, 9].

#### 2.3.3 Graph Construction

For each WSI, each cluster in its cluster set $C$ is considered to be a vertex in the vertex set $V$ of its graph representation $G = (V, E)$ [16, 17]. The geometric centre of the node is obtained as: $\mathbf{g}_c = \frac{1}{|c|} \sum_{\mathbf{p}_j \in c} \mathbf{p}_j$ whereas, the feature representation of each node is taken as the average of patch features within the cluster and denoted by $\mathbf{h}_c$. The purpose of using graphs in the proposed approach is to model the inter-relationship between neighbouring regions in the WSI to capture large-scale topology. This is done by inter-connecting nodes within a maximum connection distance of 1,500 pixels (between top left corners of patches) to construct the edge set $E$. This effectively connects all patches within 1mm in the tissue.

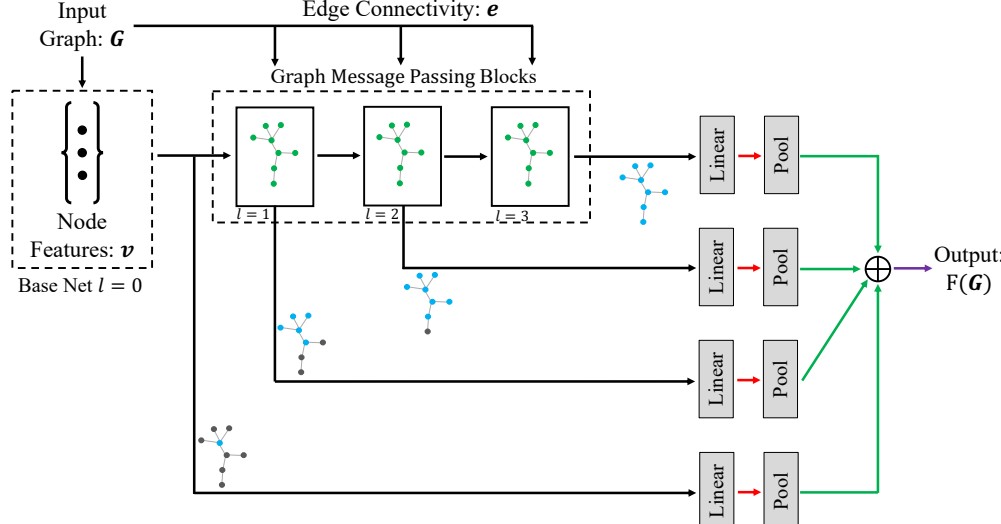

**Figure 2:** Schematic of graph neural network with message passing block layers using Edge-Conv (equation 2) or GIN-Conv layers (equation 3). The base net is composed of a linear layer, batch normalisation (BN), and gaussian error linear units activation (GELU) layer. Each message passing block includes feature level information from increasing order of neighbourhood nodes, as illustrated by the grey and blue graphs. Black lines indicate latent node representations. Red lines carry node level prediction scores. Green lines are the layer-wise WSI-level outputs which form the final WSI-level output.

## 2.4 GNN for ranking based on survival times

Following the construction of the WSI graph $G_i$, it can be passed to a graph neural network model $F(G_i; \theta)$ for prediction of WSI-level survival scores. Here, $\theta$ denotes the trainable weight parameters of the GNN. For training the GNN, we consider the graph representation $G_i, i = 1 \ldots N$ of each WSI in the training set along with the corresponding survival time $T_i$ (in days) and the event indicator variable $\delta_i \in \{0, 1\}$ representing whether the patient died of breast cancer ($\delta_i = 1$) or not ($\delta_i = 0$). We use pairwise ranking to train the GNN to predict survival scores based on the constraint that if the corresponding survival time for patient $i$ is larger than that of patient $j$ and the event for patient $j$ has taken place, i.e., $T_i > T_j | \delta_j = 1$, then GNN generated survival score for $G_i$ should be larger than that of $G_j$, i.e., $F(G_i; \theta) > F(G_j; \theta)$ [18]. This results in a pairwise comparison set $S = \{(i, j) | T_i > T_j, \delta_j = 1, i, j = 1 \ldots N\}$ consisting of WSI pairs that can be used for model training. A pairwise ranking loss is then used for model training with adaptive momentum based gradient descent as follows:

$$\theta^* = \arg\min_{\theta} \sum_{(i,j) \in S} \max(0, 1 - (F(G_i; \theta) - F(G_j; \theta))). \tag{1}$$

The survival prediction framework proposed in this work can utilise different types of graph neural network architectures. However, we have used the architecture shown in Figure-2. It is constructed from a base net consisting of a multi-layered perceptron (MLP) that operates on node level feature representations. The output of the base network is then passed to a series of graph message passing blocks (GMPBs) which take nodal connectivity in the graph into account. Each time the GMPB is invoked the architecture accumulates information from increasingly higher-order neighbours. Additionally, following each GMPB the node features are retained and passed through their own MLPs, which considers each node's embedding and the difference in embedding with its neighbours. Thus, for a given neighbourhood, $N_k$, of the node $k$ the $l^{th}$ GMPB will return the feature embedding, $h_k^{(l)}$, of a node $\mathbf{p}_k \equiv (\mathbf{g}_k, \mathbf{h}_k) \in X$, which is then passed to a linear layer generating node level predictions $f_l(\mathbf{p}_k) = \mathbf{w}_l^T \mathbf{h}_k^{(l)}$ with an appropriate activation. The resulting node level prediction

scores are then pooled, $\mathbf{F}(\mathbf{G}) = \sum_{\forall \mathbf{p} \in P} f_l(p)$ to create layer-wise WSI level scores. These layer wise scores can then be aggregated to produce an overall WSI level prediction.

It is thus the case that graph neural networks (GNN) [19], built using Edge Convolution (Edge-Conv) or graph isomorphic convolution (GIN-Conv) [20], are suitable for producing a prediction for the input graph $G$. The GNN's learning is predicated upon the ability to extract abstract representations of node level features as a function of their local neighbourhood, by simulating message passing between neighbouring nodes.

Mathematically the Edge-Conv and GIN-Conv layers can be expressed by the following functions, respectively:

$$\mathbf{h}_k^{(l)} = \sum_{u \in \mathcal{N}_k} H_\theta^{(l)}(\mathbf{h}_k^{(l-1)}, \mathbf{h}_u^{(l-1)} - \mathbf{h}_k^{(l-1)}), \tag{2}$$

$$\mathbf{h}_k^{(l)} = H_\theta^{(l)}((1 + \epsilon^{(l)}) \cdot \mathbf{h}_k^{(l-1)} + \sum_{u \in \mathcal{N}_k} \mathbf{h}_u^{(l-1)}). \tag{3}$$

Here, $\mathbf{h}_\mathbf{k}^{(\mathbf{l})}$ is the feature vector for the $k^{th}$ node on the $l^{th}$ layer (with $\mathbf{h}_\mathbf{k}^{(\mathbf{0})} = \mathbf{h}_\mathbf{k}$). As shown in equation 2, Egde-Conv works by aggregating node representation of a given node with its difference with the representation of node in its neighborhood $\mathcal{N}_k = \{u \in V | (u, k) \in E\}$ . For GIN-Conv, the first term in the right hand side of the equation 3 determines the local contributions from a given node, controlled by $\epsilon$ and the second term takes the contributions from the neighbourhood. Finally, $H_\theta^{(l)}$ is the all encapsulating multi-layer perceptron that is capable of learning the non-linear node level transformations that are required. We have experimented with both Edge-Conv and GIN-Conv layers in the paper and found that Edge-Conv offers superior predictive performance.

The GCN produces a prediction score from the summation of the node level feature representations. The MLP weights are tuned through the propagation of gradients which result from the ranking loss function discussed earlier.

## 2.5  Code and data availability

The proposed approach has been implemented in Python using the PyTorch-Geometric library for graph neural networks. The complete architecture for the model used can be found on the organisational GitHub[1] page. The graph representations, associated survival data and trained predictive models can be found on a public Google Drive[2] for download.

# 3  Experiments and Results

In order to evaluate predictive performance of the proposed approach, we have used 5 runs in each of which the dataset was randomly divided into a training and test set with 20% of the overall data for reporting test performance. We ensured that the percentage of cases with events is kept the same across training and test splits. We report the predictive performance of the proposed approach using concordance index (c-index). C-index measures the degree of concordance between relative prediction scores of test patients and their actual survival times. In line with area under the receiver operating characteristic curve (AUROC), the C-index ranges from 0.0 (inverted ranking of survival scores) to 0.5 (no concordance between predicted scores and actual survival times) to 1.0 (perfect concordance between prediction scores and actual survival times). It enables us to compare our predictive performance with previously published results. In addition to this, we also report the Kaplan-Meier survival curves for high-survival and low-survival group stratifications obtained by thresholding the prediction score generated by the model with a threshold selected using training data examples. The p-value of the log-rank test is also reported.

## 3.1  Quantitative results and Comparison

As shown in Table-1 the average concordance index over the test set for the proposed approach is $0.672 \pm 0.058$ which is markedly better in comparison to previous approaches.

---

[1]https://github.com/CCMackenzie/GNNSurvivalRankLoG
[2]https://drive.google.com/drive/folders/1BBROCRZKm5Oz3cmergZP_CemJFHkV3rx?usp=sharing

The Kaplan-Meier curve over a representative test set data split is shown in Figure-3. It shows that the prediction score generated by the proposed approach was able to produce meaningful stratification of patients into two groups whose survival is statistically significantly different ($p = 0.002$ (3 s.f) $<< 0.05$). The patients in the low group (prediction score below the threshold) have significantly better survival probability over time in comparison to patients in the high risk group. Results of this stratification were consistent over multiple evaluation runs.

We believe that the proposed method is able to perform better in comparison to other existing approaches due its effective inclusion of spatial context between the patch level features. As discussed earlier many alternative methods rely on generic grouping of random patches from the WSI. Examination of the global spatial context of micro-features in histology images is a common practice for pathologists and so exclusion of this information limits a given model's ability to perform survival prediction. We have also investigated the impact of various architectural choices which are discussed below.

### Choice of Model Layers

We have experimentally compared the predictive performance of GIN vs Edge-Conv which has resulted in a significant improvement in concordance index from $0.602 \pm 0.093$ (With GIN-Conv) to $0.672 \pm 0.058$ (with Edge-Conv).

### Role of Patch Clustering

Our analysis shows that the predictive performance in terms of test concordance index does not change much with ($0.672 \pm 0.058$) and without clustering ($0.685 \pm 0.072$). However, patch-level clustering reduces the size of the graphs leading to faster computation times. Without clustering, the average per train-test split run time is approximately 1,800 seconds whereas with clustering it stands at 770 seconds. This is, as expected, dependent upon the amount of memory available on the GPU.

### Role of Knowledge Jump Connections

We have investigated the role of "knowledge jumping" or connections from each layer to the final prediction experimentally and found that the difference between average test concordance with ($0.672 \pm 0.058$) and without ($0.687 \pm 0.074$) these connections is marginal. The addition of these connections seems to reduce the variance in prediction results.

### Choice of node level features

In addition to deep features obtained from Shuffle-Net, we have also used estimates of patch-level cellular composition i.e., counts of neoplastic, inflammatory, connective and epithelial cells in each patch, as node level features. These estimates are obtained from a machine learning method called ALBRT (see [14] for details). Using these features, the best average concordance index was $0.63 \pm 0.09$ which is lower in comparison to deep features from Shuffle-Net. We conjecture that despite lower concordances such clinical features may offer novel insights into the role of different cells in conjunction with other histologically important features such as blood vessels and mitotic figures along with genomic or transcriptomic features.

### Effect of edge connectivity threshold

We have also analyzed the impact of edge connectivity threshold used for defining the edges in the graph on test concordance figures. We have found that an optimal test concordance ($0.689 \pm 0.08$) is obtained with a connectivity threshold of 2000 pixels whereas 1500 pixels offers lower variation. The predictive performance started to drop beyond the 1mm tissue region bounds on edge connectivity between nodes. However, we believe that this range is dependent upon the choice of features and can be chosen as a design parameter for different problems.

### Impact of censoring

We have used 10 years as the censoring threshold. However, if no such artificial censoring is used the average test concordance index improves to $0.702 \pm 0.076$. This is expected due to the increase in the number of pairs of examples that contribute to the loss function in training.

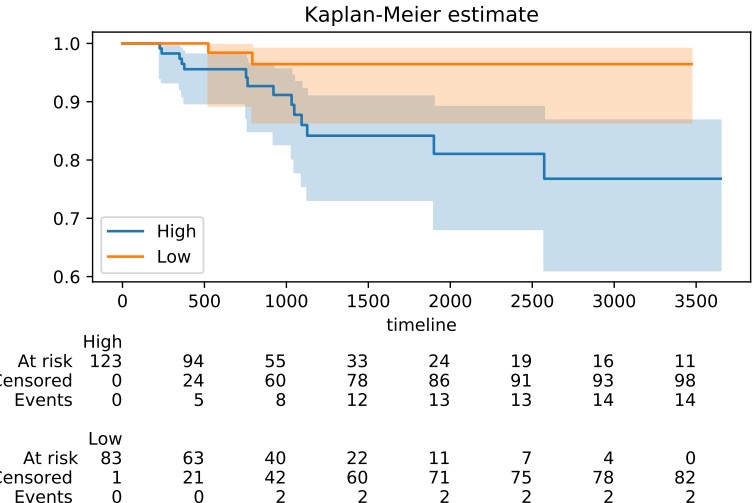

**Figure 3:** Kaplan-Meier survival curves showing the probability of disease specific survival (y-axis) over time (in days, x-axis) for high and low risk groups in the test set along with the number of patients who are at-risk, censored or experienced a survival event at various time steps (bottom).

| Method | Concordance Index $\pm$ std dev |
|---|---|
| MCAT[6] | $0.580 \pm 0.069$ |
| Attention MIL[21] | $0.564 \pm 0.050$ |
| DeepAttnMISL[4] | $0.524 \pm 0.043$ |
| Proposed (with Edge-Conv) | $0.672 \pm 0.058$ |

**Table 1:** Table of concordance indexes comparing alternative methods from elsewhere in the literature to the proposed survival model.

**Processing times**

The approach has been developed in Python using the PyTorch Geometric library for graph neural networks. On an NVIDIA RTX 3080 GPU, the per train-test split execution time is around 12 minutes (on average). The creation of graphs from node level features for the whole dataset requires (on average) 35 minutes whereas the extraction of deep features and pre-processing can be expected to take 8 to 10 minutes per whole slide image (WSI) for pre-processing, patch extraction and node level feature computation depending upon the size of tissue within the WSI.

### 3.2 Visual Results

Omission of the pooling stage when getting a prediction from the model means node level scores can be extracted to provide insight into the tumour regions that contribute most heavily towards risk. Conversion of the node prediction score to a false colour representation of each node provides a WSI visualisation in which the colour on each node corresponds to its node level prediction. Figure-4 illustrates how the proposed approach at node level, when trained for survival, can distinguish between high and low risk patients and indicate the regions that contribute most to those survival predictions. Patients selected had both experienced an event to provide a defined difference in their survival times. Node level scoring would be a useful addition to the clinical study of WSI prognostication. This visualisation capability supports the assertion that inclusion of spatial context information has great utility in machine learning applications in computational pathology.

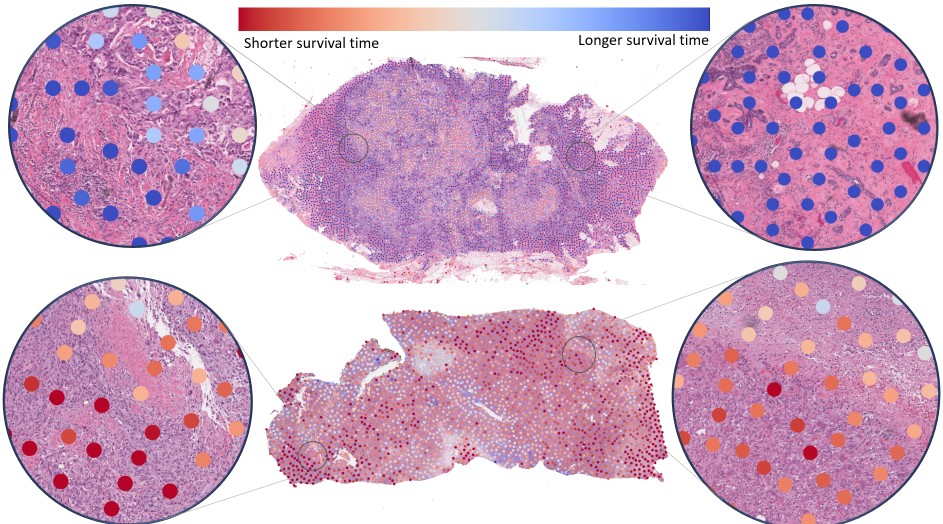

**Figure 4:** Graph visualisation of TCGA-B6-A0RS ($\delta = 1$, T = 3063 days, WSI model score = 4.515) (Top) and TCGA-AC-A2QJ ($\delta = 1$, T = 446 days, WSI model score = 1.548) (Bottom) using node level prediction scores with red to blue false colour mapping. Our initial investigation into regions associated with low survival indicates their association to larger number of pleomorphic tumour cells along with increased cellularity.

## 4   Conclusions and Future Work

In this work, we have developed a graph neural network model of whole slide images for ranking breast cancer patients based on their disease-specific survival times. The proposed approach provides a general architecture for modelling whole slide images as graphs for survival analysis and compares favourably to previous methods. The proposed approach currently relies on using deep features which are not easily explainable. It can be improved by inclusion of more clinically-oriented features so that the predictions can be explained and used for discovery of novel prognostic biomarkers. In addition to that, the correlation of the prediction scores with different clinical covariates such as age, disease subtype, grade and stage needs to be investigated. In the future, we would like to expand the analysis to other cancer types and utilise external and completely independent cohorts from other clinical centres for performance assessment. Linking histopathology based features with other genomic and transcriptomic features can further improve predictive performance.

#### Author Contribution and Acknowledgements

Callum Mackenzie: Implementation and Experimentation, Analysis, and Write-up. Muhammad Dawood: Generation of image features, and Experimentation. Simon Graham: Experiment design and planning, Paper Write-up and Review. Mark Eastwood: Code Review and Debugging, Experimentation, Analysis, and Visualisation. Fayyaz Minhas: Conceptualisation, Formulation, Write-up, and Supervision.

Fayyaz Minhas and Simon Graham are part of the PathLAKE digital pathology consortium, which is funded by the Data to Early Diagnosis and Precision Medicine strand of the government's Industrial Strategy Challenge Fund, managed and delivered by UK Research and Innovation (UKRI). Fayyaz Minhas and Mark Eastwood also acknowledge funding support from EPSRC EP/W02909X/1. Muhammad Dawood is partly funded by a PhD studentship from GlaxoSmithKline.

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
