# OpenReview forum: "Neural Graph Modelling of Whole Slide Images for Survival Ranking"
_logconference.io/LOG/2022/Conference — LoG 2022 Poster_

### Official Review · Reviewer_2tER · 2022-10-19

**Overall Score:** 6
**Confidence:** 3

**Review:**

This paper describes a graph neural network model for predicting survival time from Whole Slide Images (WSIs) in breast cancer. Due to the size of a WSI, 100, 000 × 100, 000 pixels, and the fact that the morphological features that are informative of cancer tend to occur in localised areas, existing approachs split the image into multiple small image patches and apply Multiple Instance Learning. The approaches presented here uses a GNN to model the relationship between locally close patches and to aggregate information across different patches in the graph. The architecture closely follows previous work which uses a GNN on WSIs to solve a classification problem.

Strengths:
- The paper is clearly written and easy to follow
- The experiments show good performance of this approach, substantially out-performing the baselines

Weaknesses:
- There appears to be little technical novelty. Is the model presented here equivalent to SlideGraph+ adapted to use a loss function appropriate for survival analysis?
- The authors only compare to 3 baselines - are there no other methods that could be competitive on this problem?

---

### Official Review · Reviewer_D8ex · 2022-10-21

**Overall Score:** 8
**Confidence:** 4

**Review:**

Summary:
In this work, authors propose a graph-based approach for whole-slide image (WSI) representation learning, towards the downstream task of survival prediction of breast cancer patients. Authors explicitly set the goal to achieve this purely based on WSI information, and without additional patient data (e.g. genomic or patient features). Multi-gigapixel WSIs are tiled down into 512^2 patches and deep features are extracted using a (apparently pre-trained) Shuffle-Net. Local patch clusters are formed based on spatial neighborhood and feature similarity of patches, and the cluster location and feature representation is formed by averaging of all contributing patch locations/features. Cluster centers then become graph vertices, and their feature vectors become node representations. Edge are formed by connecting patch centers with a maximum connection distance of 1500 pixels. The entire WSI is then condensed into a single feature vector representation with a custom GCN architecture: node features are processed via three layers of message passing, to aggregate neighborhood information; skip connections forward intermediate cluster/node representations; all node representations (aggregated or not) are transformed linearly and aggregated globally via sum-pooling. Authors propose a novel loss based on pair-wise survival time comparison (comparing pairs of WSIs).

Contributions: Main contributions are 1) the custom architecture, 2) the novel and problem-specific pair-wise ranking loss, 3) demonstrating the feasibility of survival prediction based on only WSI-based patient representations with high concordance.

Strong points: The paper is very well written. The condensation of multi-Gigapixel information into a single representation is an elegant application area of GNNs in medical imaging / digital pathology. It is surprisingly efficacious, too - almost surprising that it is possible  compress WSIs this strongly, while still retaining key information valuable to survical prediction, all on a WSI-level. That insight alone is a valuable contribution. The novel architecture, especially in combination with the contrastive pair-wise loss, goes beyond related works like SlideGraph. Methodologically, I see no red flags in the paper, the experiments performed are not very varied, but executed in a sound manner, with 5 runs of random train/val/test splits of WSIs, from a sizable subset of nearly 1000 subjects from the TCGA repository. Results are very convincing, in terms of concordance indices compared to SOTA, and Kaplan-Meier survival curves of high- and low-risk groups, which indicates a potential value of the proposed methods for clinical survival prediction.

Pointers for improvement:
The concordance index of the proposed method, as the authors denote, is markedly better than previous approaches (0.712 vs 0.580/0.564/0.524 by competing methods). It would have been interesting though to perform an ablation study. As it stands, it is unclear where this drastic improvement is coming from exactly. Is the novel architecture, and the WSI-global cluster/node feature aggregation responsible, is it the skip connections, or is it the pairwise ranking loss? In a follow-up study I would recommend analyzing this in more detail.

Given the clarity of the manuscript, the intuitive (yet still surprisingly effective) architecture of the model, and the convincing results, I recommend acceptance of the paper. There are no real concern otherwise, the only suggestion is for extension of the work in follow-up manuscripts, and conducting of ablation studies.

Additional suggestions for improvement:
- It would be interesting to report computation requirements: hardware used, training / inference times, libraries used (the announced release of source code will clarify most of these, but mentioning in the paper would be helpful).
- Figure 4: please explain better in the figure caption what the WSIs and what the enlarged sections are depicting, especially for non-domain experts. Do false colors coincide with expert knowledge of pathologists?

The paper meets requirements for a long paper.

---

### Official Review · Reviewer_kVhH · 2022-10-22

**Overall Score:** 6
**Confidence:** 3

**Review:**

This paper first proposes to apply graph neural networks on whole slide images for survival analysis, where spatially-localized features along with their context are encoded with GNNs for survival score prediction. Experiment results and visualizations demonstrate its superiority compared with limited existing methods.

Pros:
- The image preprocessing, experiment setup, and evaluation are well clarified.
- The graph visualization demonstrates that the node level score and spatial context information can effectively distinguish between high and low-risk patients that contribute most to survival predictions.
- The patch clustering can potentially provide interpretability for identifying tumor regions.

Cons:
- The benefit of modeling patch clusters as a graph is not well-argued or supported by experimental results. The inter-relationship between neighboring regions does sound fancy and promising in capturing higher-level information, while at the same time, the global pooling for producing the graph-level representation and the clustering step could also lead to information losses, compared with using the original full image patches. The necessity of using graphs to model WSI should be more clearly stressed for survival analysis.
- Limited exploration of the model designs: the backbone GNN model is simply GIN. The authors didn't vary the backbone much or adapt the model architecture design, while the GIN model is pretty simple and it is essential to take the average of neighborhoods. This may further introduce the over-smoothing issue, which we do not expect to happen across clusters. The authors are suggested to try different types of backbone and give a comparison among them.
- Lack of explanation for spatial clustering: is there any data uniqueness for the WSI data that motivates the design of spatial clustering? Why should we cluster different patches? How does the patch feature extraction influence the clustering results?
- The proposed approach relies only on deep features. More clinical-oriented features such as genomic and transcriptomic ones can be much more helpful and meaningful compared with the geometric features derived from GNNs.
- How is the threshold (1,500 ) chosen for the maximum connection distance of 1,500 pixels when deciding on edges?

---

### Official Review · Reviewer_vBHU · 2022-11-06

**Overall Score:** 6
**Confidence:** 5

**Review:**

The manuscript entitled “Neural Graph Modelling of Whole Slide Images for Survival Analysis” proposed a GNN-based survival ranking framework for analyzing whole slide images and performing survival prediction. The proposed model is validated and evaluated on a public WSI dataset from TCGA and shows superior performance to several other deep learning methods.

Clarification of the methodology description needs to be improved, specifically:

•	The description for graph edge construction: “This is done by inter-connecting nodes within a maximum connection distance of 1,500 pixels to construct the edge set E.” is ambiguous: is the maximum distance defined by the distance between boundaries of the nodes or the geometric center of the nodes? For the former case, the distance measurement might be affected by the relative spatial positioning and the size of the clusters; for the latter case, if a huge node is formed (i.e., with a radius of >1,500 pixels), it will be not connecting to any other nodes by definition.
•	The index “j” is used without definition in the sentence, “We use the pairwise ranking to train the GNN to predict survival scores based on the constraint that if the corresponding survival time for i is larger than that of j and the event for j has taken place…” in section 2.4. It seems that the sentence “We construct a pairwise comparison set S = {(i, j)|Ti > Tj , δj = 1, i, j = 1 . . .N} consisting of WSI pairs that can be used for model training” shall precede it.

---

### Meta-Review · Area_Chair_aCAt · 2022-11-16

**Confidence:** 4
**Recommendation:** Accept

**Meta Review:**

This paper falls into the category of graph representation learning frameworks for digital pathology. The authors capture whole slide images (WSIs) with a graph and propose a GNN based method for pairwise ranking between WSIs based on the survival score, which leads to survival prediction of breast cancer patients. All reviewers agree that the paper is interesting and generally well-written. Although GNNs have been extensively used for modelling WSIs, this paper provides an additional contribution in the way that the graph is constructed (i.e, clusters are nodes), and the new and problem-specific pair-wise ranking loss. Experimental results are convincing. Based on this evaluations, and my own reading of the paper, I recommend acceptance of the manuscript.

---

### Decision · Program_Chairs · 2022-11-23

Accept (Poster)